# Data-Driven Analysis of High-Temperature Fluorocarbon Plasma for Semiconductor Processing

**DOI:** 10.3390/s24227307

**Published:** 2024-11-15

**Authors:** Sung Kyu Jang, Woosung Lee, Ga In Choi, Jihun Kim, Minji Kang, Seongho Kim, Jong Hyun Choi, Seul-Gi Kim, Seoung-Ki Lee, Hyeong-U Kim, Hyeongkeun Kim

**Affiliations:** 1Electronic Convergence Material and Device Research Center, Korea Electronics Technology Institute (KETI), Seongnam 13509, Republic of Korea; skjang@keti.re.kr (S.K.J.); wslee@keti.re.kr (W.L.); gana034@keti.re.kr (G.I.C.); kjh96@keti.re.kr (J.K.); jhchoi@keti.re.kr (J.H.C.); seulgi11.kim@keti.re.kr (S.-G.K.); 2Semiconductor Manufacturing Research Center, Korea Institute of Machinery and Materials (KIMM), Daejeon 34103, Republic of Korea; kmj0302@kimm.re.kr (M.K.); kimho6344@kimm.re.kr (S.K.); 3Department of Materials Science and Engineering, Chungnam National University (CNU), Daejeon 34134, Republic of Korea; 4Department of Materials Science and Engineering, Pusan National University (PNU), Busan 46241, Republic of Korea; ifriend@pusan.ac.kr; 5Nano-Mechatronics, KIMM Campus, University of Science & Technology (UST), Daejeon 34113, Republic of Korea

**Keywords:** fluorine-based plasma, amorphous carbon layer, gas temperature, time-of-flight mass spectrometry (ToF-MS), principal component analysis (PCA), non-negative matrix factorization (NMF), first-order plus dead time (FOPDT) model, process optimization

## Abstract

The semiconductor industry increasingly relies on high aspect ratio etching facilitated by Amorphous Carbon Layer (ACL) masks for advanced 3D-NAND and DRAM technologies. However, carbon contamination in ACL deposition chambers necessitates effective fluorine-based plasma cleaning. This study employs a high-temperature inductively coupled plasma (ICP) system and Time-of-Flight Mass Spectrometry (ToF-MS) to analyze gas species variations under different process conditions. We applied Principal Component Analysis (PCA) and Non-negative Matrix Factorization (NMF) to identify key gas species, and used the First-Order Plus Dead Time (FOPDT) model to quantify dynamic changes in gas signals. Our analysis revealed the formation of ${\mathrm{COF}}_{3}$ at high gas temperatures and plasma power levels, indicating the presence of additional reaction pathways under these conditions. This study provides a comprehensive understanding of high-temperature plasma interactions and suggests new strategies for optimizing ACL processes in semiconductor manufacturing.

## 1. Introduction

The rapid advancement of the semiconductor industry has necessitated increasingly complex and delicate manufacturing processes, particularly in the etching processes of 3D-NAND and DRAM technologies. High aspect ratio etching technology has become crucial, with the Amorphous Carbon Layer (ACL) widely utilized as a hard mask due to its excellent etch resistance [1,2]. This enables fine patterning of high aspect ratio structures, with ACL typically deposited using Plasma Enhanced Chemical Vapor Deposition (PECVD) at temperatures exceeding 400 °C [3,4,5,6]. The deposition process adjusts the sp^2^ and sp^3^ bond ratios by changing the temperature to control material properties, such as etch resistance, and can be conducted at temperatures up to 650 °C [7,8].

However, ACL deposition chambers are prone to carbon particle contamination, requiring periodic cleaning with fluorine-based plasmas to maintain uniform process conditions. During high-temperature plasma deposition and cleaning processes, the equipment components face extreme thermal and chemical stresses, demanding high performance and durability from the materials used for the inner walls of ACL deposition chambers and high-temperature ceramic heaters [9]. Despite the significance of high-temperature conditions in ACL deposition and chamber cleaning processes, a comprehensive evaluation of plasma environment changes under high temperatures has not been conducted. Existing research on high-temperature gas plasmas has primarily focused on the study of magnetogasdynamics at ultra-high temperatures [10,11] or the condensation and growth of materials [12], with little emphasis on semiconductor processing applications. Only a limited number of simulation studies have addressed the behavior of semiconductor process gases under such conditions [13]. Although techniques such as electron density measurements using resonant probes in high-temperature gases have been explored [14,15,16], these methods have limitations when applied to complex plasma chemistries involving etching gases. Specifically, in environments where reactive radicals are present, conventional diagnostics struggle to accurately capture the density and distribution of these species due to the high reactivity and short-lived nature of radicals. Our research addresses this gap by providing experimental data that directly measure the changes occurring in real semiconductor process environments, offering insights that can be applied to optimize plasma processes for high-temperature applications.

The absence of experimental data on high-temperature plasma behavior, particularly in ACL deposition and cleaning processes, presents a challenge for process development. The complex interactions between plasma characteristics and process variables cannot be fully understood, making process optimization reliant on statistical approaches that treat the plasma chamber as a black box. This limitation hinders the ability to effectively optimize and control ACL processes in different equipment or operating environments. Therefore, obtaining precise measurements of internal plasma variables, such as radicals and gas species densities, in response to changes in process conditions is crucial. Such data are necessary for achieving more effective and adaptable process development in ACL applications.

Among these parameters, monitoring fluorine (F) is particularly important due to its role as a highly reactive etching agent widely used in the semiconductor industry [17]. However, understanding the generation and behavior of F radicals in the etching plasma chamber remains challenging. The quantitative analysis of fluorine is particularly difficult due to its high reactivity, elevated ionization potential (17.4 eV), and the lack of prominent spectral lines in the visible spectrum [18]. Furthermore, its low ionization efficiency in mass spectrometric systems results in weak signal intensities, making accurate detection even more complex [19,20].

The transient nature of F radicals presents additional obstacles for precise quantification and real-time monitoring. Due to their high reactivity, F radicals rapidly recombine, interact with other plasma species, or collide with the chamber walls, resulting in a substantial reduction in their concentration within tens of milliseconds [21,22]. This rapid decay continues as the gas flows through the interface between the processing chamber and the measurement equipment, continuously depleting the F-radical concentration. As a result, accurately measuring and detecting F radicals in real-time remains a significant challenge for process control and optimization.

To overcome these limitations, we leveraged the advantage of Time-of-Flight Mass Spectrometry (ToF-MS), which allows for the parallel collection of signals from all gas species. To effectively analyze the collected signals, we applied Principal Component Analysis (PCA) to distinguish key elements, focusing on the variations in fluorine-containing species to estimate the F-radical concentration. By employing this methodology, we obtained a comprehensive profile of the plasma environment and accurately described the interactions and dynamics of fluorine–carbon species under various temperature conditions. For this purpose, we developed a high-temperature inductively coupled plasma (ICP) system integrated with ToF-MS. This analytical approach not only reveals the complex dynamics of F radicals but also provides insights that can significantly enhance process optimization and control strategies.

Through additional PCA and clustering of the selected key gas species, we visualized the complex multidimensional data and identified key variables and their interactions. This method provided a clearer understanding of the plasma process environment and enabled the effective interpretation of fluorine-containing species’ behavior.

## 2. Experimental Setup and Data Collection

Figure 1 illustrates a schematic diagram of the entire system used for collecting and analyzing gas species data based on process variables. A gas mixture of ${\mathrm{CF}}_{4}$/$O_{2}$/Ar at a 2:2:1 ratio was introduced into a vacuum chamber to create a fluorocarbon mixed gas environment, with the process pressure maintained at 2 mTorr. To generate plasma, a 13.56 MHz power source was supplied to an ICP coil, and a hot-wall heater was employed to heat the chamber for a high-temperature gas environment. To ensure the formation of plasma at the appropriate position within the chamber, an electrically grounded chuck was installed inside the heater via vacuum feedthroughs. Downstream of the chamber, a gas collection line was branched off from the main pump line and connected to the ToF-MS system. This gas sampling line was maintained at 100 °C using heating tape to prevent condensation or deposition of gas species. Inside the ToF-MS, dry pumps were utilized to draw in the process gas from the chamber. Gas species quantities were measured using ToF-MS while the ICP plasma power varied from 100 W to 900 W at different temperature conditions (20 °C, 200 °C, 400 °C, and 600 °C). Each experiment was conducted after the chamber temperature had sufficiently stabilized to match the set experimental conditions. ToF-MS ionizes gases through electron impact and then accelerates the ionized gases using an electric field, allowing for the parallel measurement of each gas species based on their mass-to-charge ratio.

This principle allows ToF-MS to accurately measure the relative amounts of different gas species during the process. Changes in ToF-MS signals over time, following the application of plasma power, were analyzed using a First-Order Plus Dead Time (FOPDT) model and regression analysis to calculate the variations in each gas species. This method provides precise insights into the dynamic behavior of gas species in response to changes in process conditions.

## 3. Data Analysis Methodology

### 3.1. First-Order Plus Dead Time (FOPDT) Model

The FOPDT model is widely used in process control and systems analysis to represent the dynamic behavior of systems [23,24]. This model is particularly useful for understanding the time-dependent changes in process variables, such as gas species signal intensities, in response to changes in input variables such as power or temperature.

The FOPDT model captures the essential dynamics of a system using three main parameters: gain (*K*), time constant (τ), and dead time (θ). Gain (*K*) represents the system’s steady-state response to a unit change in input. The time constant (τ) indicates how quickly the system responds to changes, defining the speed of the system’s response. Dead time (θ) is the delay between an input change and the start of the system’s response. The standard form of the FOPDT model is given by the following transfer function:(1)$$G\left(s\right)=\frac{Y(s)}{X(s)}=\frac{Ke^{-\theta s}}{\tau s+1}$$
where G(s) is the transfer function of the system, Y(s) is the Laplace transform of the output, X(s) is the Laplace transform of the input, and *s* is the complex frequency variable. In our study, we maintained a constant temperature and gas flow while applying a set ICP plasma power, thus using a unit step function to represent the changes in the gas species. Given that the system’s input X(s) is a unit step function, by applying the inverse Laplace transform to the transfer function, we obtain the time-domain equation for the FOPDT model:(2)$$y\left(t\right)=K\left(1-e^{-\frac{t-\theta }{\tau }}\right)u\left(t-\theta \right)$$
where y(t) is the system output (ToF-MS signal intensity), u(t−θ) is the step input function delayed by the dead time θ, and *t* is time. The model simplifies complex dynamics into key parameters, making it easier to interpret the system’s behavior without the need for overly complex models.

The FOPDT model is particularly suitable for analyzing changes in ICP plasma power before and after application due to its ability to account for gain (*K*), which quantifies the sensitivity of ToF-MS signals to power changes, simplifying complex dynamics into key parameters for easier analysis. Additionally, the model accounts for dead time (θ), which represents the delay between power application and the observable ToF-MS response, and the time constant (τ), which indicates how quickly the ToF-MS signal stabilizes. However, in our experiment, the gain (*K*) was of primary importance for quantitative analysis. Dead time (θ) and time constant (τ) were considered in the regression fitting process due to potential errors from unavoidable dead volumes in the interface between the plasma chamber and the ToF-MS system.

To apply the FOPDT model, follow these steps: Record the ToF-MS signals of gas species intensities over time while maintaining a set ICP power, constant temperature, and stable gas flow. Identify the initial response time and the steady-state change in gas species signal intensities. Utilize regression analysis with the time-domain equation for the FOPDT model to estimate the parameters (*K*, τ, and θ). Validate the model by comparing its predicted response to the actual ToF-MS data. This method allowed for a precise understanding of the dynamic behavior of gas species signal intensities in response to changes in ICP plasma power, demonstrating robustness even in the presence of dead volume.

### 3.2. Principal Component Analysis (PCA)

PCA is a statistical method used for reducing the dimensionality of large datasets while preserving the variability present in the data. This technique transforms original variables into a new set of uncorrelated variables called principal components, ordered by the amount of variance they capture from the data [25]. PCA is particularly useful in simplifying complex datasets and highlighting their underlying structure.

In our experiment, PCA was applied for two main purposes using data collected from ToF-MS. Firstly, we used the absolute values of the principal component loadings to select the key gas species. This method is effective in identifying the primary gas species that contribute the most to the overall variability of the data, significantly reducing the data volume and consequently lowering monitoring costs [26]. Moreover, this approach offers flexibility for application to other processes as well. Secondly, PCA was employed to visualize the multidimensional process gain variable, which includes information on different gases, in a 2D plane. This visualization aids in understanding the relationships and variations between different gas species under various temperature and ICP plasma power conditions. By reducing the dataset’s complexity and focusing on the most influential variables, PCA enables a more efficient and insightful analysis of the chemical reactions occurring in the plasma.

We utilized ToF-MS data collected in parallel to identify key signals using the absolute value of the principal component loadings. The dataset is structured such that the columns represent different gas signals collected based on their molecular weights (*m*) and the rows represent time points. This forms a data matrix *X*:(3)$$X=\left[\begin{array}{cccc} x_{11} & x_{12} & \cdots & x_{1M}\\ x_{21} & x_{22} & \cdots & x_{2M}\\ \vdots & \vdots & \ddots & \vdots \\ x_{T1} & x_{T2} & \cdots & x_{TM} \end{array}\right]$$
where *T* is the number of time points and *M* is the total number of gas signals. We performed PCA without standardization because each column represents signal strength for different gases, with consistent units and ranges. The absolute magnitude of signal strength reflects the amount of chemical reaction, and standardizing would obscure important differences between gases and exaggerate differences for those that did not participate in the chemical reactions. Thus, not standardizing preserves these significant inherent differences. The covariance matrix *C* is calculated as follows:(4)$$C=\frac{1}{T-1}X^{T}X$$

The covariance matrix *C* captures the linear relationships between the gas species. The eigenvalues and eigenvectors of *C* are then calculated. The eigenvectors ($v_{i}$) represent the principal components, while the eigenvalues ($\lambda_{i}$) indicate the amount of variance explained by each principal component. These are obtained by solving the characteristic equation:(5)$${\mathrm{Cv}}_{i}=\lambda_{i}v_{i}$$
where
(6)$$v_{i}={\left[\begin{array}{cccccc} v_{1i} & v_{2i} & \cdots & v_{mi} & \cdots & v_{Mi} \end{array}\right]}^{T}$$

The principal components are ranked based on their eigenvalues, and the top components that explain the majority of the variance are selected. The absolute values of the principal component loadings are used to identify the most significant variables. The contribution of the *m*-th gas species signal to the *i*-th principal component was given by the absolute value of the loading $|l_{mi}|$. The loadings are related to the eigenvectors by the equation
(7)$$|l_{mi}\left|=\right|v_{mi}|\sqrt{\lambda_{i}}$$
where $v_{mi}$ is the *m*-th species of the *i*-th eigenvector and $\lambda_{i}$ is the *i*-th eigenvalue. To define the overall importance $I_{m}$ of each gas species, we sum the contributions of that signal across all principal components. This is given by
(8)$$I_{m}^{PCA}=\sum_{i=1}^{k}|l_{mi}|=\sum_{i=1}^{k}\left|v_{mi}\right|\sqrt{\lambda_{i}}$$
where $I_{m}^{PCA}$ is the importance of the *m*-th gas species, *i* is the index of the principal components, *k* is the total number of principal components considered, and $\lambda_{i}$ is the *i*-th eigenvalue. The relative PCA importance $R_{m}^{PCA}$ of each gas species is then defined as the importance of the gas species divided by the sum of the importances of all gas species:(9)$$R_{m}^{PCA}=\frac{I_{m}^{PCA}}{\sum_{m=1}^{M}I_{m}^{PCA}}$$

This analysis provides insights into the key gas species that contribute the most to the overall variance in the data, enabling a more focused and effective analysis of the plasma process. In the previous section, we calculated the gain (*K*) for each gas using the FOPDT model to effectively analyze variations in key process gases under different temperature and ICP power conditions. To visualize and analyze these gain values in a 2D plane, we again performed PCA, where each column represents a key gas and each row corresponds to a process variable. Following the above steps, we calculated the covariance matrix, eigenvalues, and eigenvectors.

The principal component loadings are examined to identify significant factors affecting the process. The results help optimize the plasma process by focusing on the key parameters identified by PCA. The two principal components that explained the most variance were selected for 2D visualization and analyzed to interpret the underlying patterns and relationships among the process parameters. Applying PCA to the plasma process data effectively reduced the dimensionality of the dataset while preserving the most important information. This analysis provides insight into the critical parameters that influence process results, enabling better control and optimization of plasma processing.

### 3.3. Non-Negative Matrix Factorization (NMF)

NMF is another dimensionality reduction technique, particularly useful for interpreting complex datasets with non-negative constraints [27]. Unlike PCA, which uses eigenvectors and eigenvalues to reduce dimensionality, NMF decomposes the original matrix into two lower-dimensional matrices with non-negative elements.

Given the data matrix X, NMF seeks to approximate it by the product of two matrices, W and H:(10)X≈WH

Here, W is a t×k matrix representing the basis vectors and H is a k×m matrix representing the coefficients. The factorization aims to minimize the reconstruction error, often measured by the Frobenius norm:(11)$$\min_{W,H}{∥X-WH∥}_{F}$$
where
(12)$${∥X-WH∥}_{F}=\sqrt{\sum_{i=1}^{T}\sum_{j=1}^{M}{\left(x_{ij}-{\left(WH\right)}_{ij}\right)}^{2}}$$

The NMF process begins with the initialization of W and H with non-negative values. Iteratively, W and H are updated to reduce the reconstruction error using the following multiplicative update rules:(13)$$H\leftarrow H\circ \frac{W^{T}X}{W^{T}WH}$$
(14)$$W\leftarrow W\circ \frac{XH^{T}}{WHH^{T}}$$

Here, ∘ denotes element-wise multiplication, and the divisions are also element-wise. These updates are repeated until convergence, typically when the change in reconstruction error falls below a threshold.

The basis vectors in W correspond to the key signals (similar to principal components in PCA), and the coefficients in H represent the contributions of these key signals to the original data. The matrix H captures how much each basis vector (key signal) contributes to each original gas signal, reflecting the relative importance of the gas species.

In the fluorocarbon plasma process, the gases entering the chamber were maintained at a constant level. When ICP plasma power is applied to the chamber, the gases interact and undergo chemical reactions, resulting in changes in the amounts of existing gas species or the creation of new ones. These changes are monitored by ToF-MS, and the data obtained are inherently non-negative, making them particularly suitable for NMF without the need of additional preprocessing.

In the NMF framework, gas species whose abundances change significantly due to plasma interactions have larger coefficients in the matrix H. The basis vectors in W represent underlying gas behavior patterns, and the coefficients in H indicate the strength of each gas signal’s contribution to these patterns. Consequently, gas species exhibiting greater alterations are identified by larger coefficients in H, thereby emphasizing their significance. To ascertain the importance of each gas species in NMF $I_{m}^{\mathrm{NMF}}$, the contribution of each gas signal is summed across all basis vectors:(15)$$I_{m}^{\mathrm{NMF}}=\sum_{i=1}^{k}h_{im}$$
where $h_{im}$ is the element of matrix H corresponding to the contribution of the *i*-th basis vector to the *m*-th gas signal. The relative NMF importance $R_{m}^{\mathrm{NMF}}$ of each gas species is then defined as the importance of the gas species divided by the sum of the importances of all gas species:(16)$$R_{m}^{\mathrm{NMF}}=\frac{I_{m}^{\mathrm{NMF}}}{\sum_{m=1}^{M}I_{m}^{\mathrm{NMF}}}$$

This is analogous to the relative PCA importance $R_{m}^{PCA}$. By analyzing these contributions, we can identify the most significant gas species and their relative importance, providing insights into which gas species are most affected by plasma reactions and are key to understanding and optimizing the plasma process.

## 4. Results and Discussion

### 4.1. ToF-MS and Multivariate Methods for Key Gas Species Selection in ${\mathrm{CF}}_{4}$ Mixed Plasma

Figure 2 shows an example of data collected over time using ToF-MS. To collect the data, a ${\mathrm{CF}}_{4}$ mixed gas was injected into the chamber, and a pressure of 2 mTorr and a target temperature were maintained throughout the entire process. At the same time, the gas species in the chamber were monitored by ToF-MS. After a sufficient amount of steady-state ToF-MS data had been obtained, an ICP plasma power of 500 W was applied for 600 s. Some molecular weight data as a function of time are shown in Figure 2a. The raw data contain considerable noise and insignificant signals, so it is necessary to distinguish significant signals.

To accomplish this, PCA and NMF were applied to analyze the importance of each gas species by molecular weight, as shown in Figure 2b. Gases involved in the plasma reaction show considerable variation in quantity as the plasma is turned on and off, resulting in increased variance. PCA and NMF can be used to emphasize these significant gas species while suppressing irrelevant noise signals with less variance. Some gas species that were difficult to detect in the cumulative time data (green) were identified using PCA importance (blue) and NMF importance (orange).

In ToF-MS, various factors can influence signal accuracy, such as the dependence of flight time on the starting position of ions, spatial gradients in the accelerating field, and potential timing errors caused by rapid changes in the accelerating pulse [28,29]. These factors can introduce inaccuracies that propagate through subsequent analyses, leading to erroneous interpretations. To mitigate these inaccuracies and focus on the most relevant gas species, we selected only the species corresponding to molecular weights that exhibit local maxima in importance.

Table 1 presents the relative importance values obtained from PCA and NMF for various molecular weights (*m*/*z*). This table lists gas species with local maximum importance values—those having higher importance compared to their neighboring species—and orders them by descending relative importance. Although the relative importance values obtained from PCA and NMF are generally similar, some differences in the rankings and magnitudes are observed due to the distinct characteristics of the two methods.

PCA captures the global variance structure of the data and allows for both positive and negative loading values, enabling the representation of opposing relationships between variables. This characteristic is particularly valuable in plasma chemistry, as it allows PCA to capture conflicting relationships between variables, such as the production and consumption of chemical species due to secondary reactions within the plasma chamber. In contrast, NMF emphasizes non-negative, additive parts-based representations, focusing on capturing dominant features present in the data [30], which can be especially effective for the non-negative ToF-MS measurements. However, because the components in NMF are not orthogonal, there can be redundancy or overlap among variables, potentially leading to certain chemical species being unnecessarily emphasized multiple times across different components. Despite these methodological differences, the top eight chemical species identified by both methods were identical, and the ninth and tenth most important species in PCA were ranked tenth and fourteenth in NMF, respectively. This similarity indicates that both methods consistently identify the key gas species involved in the plasma processes.

In selecting the most significant gas species, those with relative importance values exceeding 0.1% in both PCA and NMF analyses were considered. Specifically, the mass-to-charge ratio (*m*/*z*) of 85 corresponds to ${\mathrm{COF}}_{3}$, a fluorinated compound containing fluorine, which is crucial in plasma chemistry. Notably, ${\mathrm{COF}}_{3}$ showed a high correlation with ${\mathrm{COF}}_{2}$ and COF at elevated temperatures, indicating that ${\mathrm{COF}}_{3}$ participates in reactions at high temperatures. Similarly, the *m*/*z* of 119 likely represents $C_{2}F_{5}$. This species exhibited a weak negative correlation with the input gases (${\mathrm{CF}}_{x}$, $O_{2}$) and a positive correlation with other reaction products, suggesting that $C_{2}F_{5}$ is formed within the plasma chamber through reactions of the input gases, indicating active plasma chemistry generating new fluorinated species. On the other hand, *m*/*z* 78 (estimated as $C_{2}F_{5}O$) showed relative importance values above 0.1% in both PCA and NMF, but did not show a clear correlation with the other species, so we excluded it from the analysis. By combining the relative importance values from both methods with chemical relevance and correlation analysis, the ten gas species discussed in the manuscript were selected as the most significant for the study.

Based on these findings, Figure 2c illustrates the ion count over time for the ten most significant gas species, as identified by PCA. In ${\mathrm{CF}}_{4}$ mixed plasma, fluorocarbons and fluorinated compounds, such as ${\mathrm{CF}}_{3}$, ${\mathrm{CF}}_{2}$, and $C_{2}F_{5}$ (molecular weights 69, 50, and 119, respectively), arise from ${\mathrm{CF}}_{4}$ dissociation. Along with this dissociation, ${\mathrm{CF}}_{4}$ is not detected because it breaks down into ${\mathrm{CF}}_{3}$ during the electron impact ionization process in mass spectrometry [31,32]. The oxidation of carbon results in the formation of carbon oxides, including ${\mathrm{CO}}_{2}$ and CO (molecular weights 44 and 28, respectively), result from oxidation processes. Carbonyl fluorides, such as COF, ${\mathrm{COF}}_{2}$, and ${\mathrm{COF}}_{3}$ (molecular weights 47, 66, and 85, respectively), are intermediates formed through interactions between carbon, oxygen, and fluorine species in the plasma. Ar (molecular weight 40) serves as an inert carrier gas. Conversely, most signals that remained consistent upon ICP plasma power application were diminished.

### 4.2. Calculation of F-Radical Production and FOPDT Model Analysis

In semiconductor etching processes, particularly those involving silicon-based materials, fluorine radicals (F radicals) are crucial due to their high reactivity. F radicals efficiently break silicon bonds and form volatile silicon fluoride compounds, enabling precise etching. In the context of advanced 3D-NAND and DRAM technologies, maintaining the integrity of the ACL masks and managing carbon particle contamination in ACL deposition chambers are critical [7,33]. This necessitates the use of high-temperature gas plasma processes and fluorine-based plasma cleaning.

To optimize these processes, it is essential to understand the production and behavior of F radicals under high-temperature conditions. However, F radicals have a limited lifetime in the gaseous molecular state due to their high reactivity, high electronegativity, and high ionization potential, making them difficult to collect and analyze. To address this issue, we propose a method that utilizes the signals of other species participating in reactions involving F to perform calculations. This is feasible due to the fact that ToF-MS gathers all gaseous species in parallel. The resulting data matrix, denoted as $X_{r}$, is structured based on the species selected using $R_{m}^{\mathrm{PCA}}$. This matrix captures the time evolution of the aforementioned species and serves as the foundation for calculating the total F radicals produced during the process. The selected species are ${\mathrm{CF}}_{3}$, ${\mathrm{CF}}_{2}$, $C_{2}F_{5}$, Ar, $O_{2}$, ${\mathrm{CO}}_{2}$, CO, COF, ${\mathrm{COF}}_{2}$, and ${\mathrm{COF}}_{3}$. The reduced data matrix $X_{r}$ is as follows:(17)$$X_{r}=\left[\begin{array}{cccccccccc} x_{1\;{\mathrm{CF}}_{3}} & x_{1\;{\mathrm{CF}}_{2}} & x_{1\;C_{2}F_{5}} & x_{1\;\mathrm{Ar}} & x_{1\;O_{2}} & x_{1\;{\mathrm{CO}}_{2}} & x_{1\;\mathrm{CO}} & x_{1\;\mathrm{COF}} & x_{1\;{\mathrm{COF}}_{2}} & x_{1\;{\mathrm{COF}}_{3}}\\ x_{2\;{\mathrm{CF}}_{3}} & x_{2\;{\mathrm{CF}}_{2}} & x_{2\;C_{2}F_{5}} & x_{2\;\mathrm{Ar}} & x_{2\;O_{2}} & x_{2\;{\mathrm{CO}}_{2}} & x_{2\;\mathrm{CO}} & x_{2\;\mathrm{COF}} & x_{2\;{\mathrm{COF}}_{2}} & x_{2\;{\mathrm{COF}}_{3}}\\ \vdots & \vdots & \vdots & \vdots & \vdots & \vdots & \vdots & \vdots & \vdots & \vdots \\ x_{T\;{\mathrm{CF}}_{3}} & x_{T\;{\mathrm{CF}}_{2}} & x_{T\;C_{2}F_{5}} & x_{T\;\mathrm{Ar}} & x_{T\;O_{2}} & x_{T\;{\mathrm{CO}}_{2}} & x_{T\;\mathrm{CO}} & x_{T\;\mathrm{COF}} & x_{T\;{\mathrm{COF}}_{2}} & x_{T\;{\mathrm{COF}}_{3}} \end{array}\right]$$

The contribution of each species to the production of F radicals is represented by the vector $F_{c}$, which quantifies the number of F radicals produced per molecule of each species. The vector $F_{c}$ is defined as
(18)$$F_{c}={\left[\begin{array}{cccccccccc} -3 & -2 & -5 & 0 & 0 & 0 & 0 & -1 & -2 & -3 \end{array}\right]}^{T}$$

In this ${\mathrm{CF}}_{4}$ mixed plasma, a reduction in the concentration of ${\mathrm{CF}}_{3}$ results in the generation of three times the quantity of F radicals. Therefore, ${\mathrm{CF}}_{3}$ contributes −3 F radicals, ${\mathrm{CF}}_{2}$ contributes −2 F radicals, and $C_{2}F_{5}$ contributes −5 F radicals. Ar, $O_{2}$, ${\mathrm{CO}}_{2}$, and CO do not contribute any F radicals, as they are either inert or do not contain fluorine. COF, ${\mathrm{COF}}_{2}$, and ${\mathrm{COF}}_{3}$ contribute −1, −2, and −3 F radicals, respectively.

The total F radicals produced ($F_{p}$) can be calculated by multiplying the matrices $X_{r}$ and $F_{c}$:(19)$$F_{p}=X_{r}F_{c}$$Species that were not selected for $X_{r}$ either did not participate significantly in the reactions or had negligible detectable concentrations; thus, they were excluded from the calculations.

### 4.3. Dynamic Analysis of Gas Species Transitions Using FOPDT Modeling

Figure 3 illustrates the time-dependent changes in the signals of selected gas species (${\mathrm{CF}}_{3}$, $O_{2}$, and ${\mathrm{CO}}_{2}$) measured using ToF-MS during the plasma process. Due to the presence of a dead volume in the connection interface between the process chamber and the monitoring equipment, delays may occur, potentially leading to signal distortion. The FOPDT model corrects these delays, thereby ensuring that the observed concentration changes reflect the plasma process itself and not artifacts caused by the physical structure of the chamber.

The FOPDT model divides the dynamic behavior of the system into two components: dead time and first-order response. The dead time is used to describe delays that are due to external factors, such as the physical configuration of the system and the measurement equipment being used. The first-order response, defined by the process gain and time constant, uses differential equations to model the intrinsic kinetics of the process. Despite the rapidity of plasma chemical reactions, the migration delay of gas between the main chamber and the monitoring equipment can result in concentration discrepancies. Given that gas concentrations fluctuate due to diffusion and drift, which are represented by differential equations, it is appropriate to utilize a model based on differential equations. By applying an FOPDT model, this complex gas transfer process can be simplified, and the process gain reflects the dynamics of the plasma process in the main chamber, independent of various delays [24]. This approach ensures that the observed concentration changes are representative of the plasma process without distortions due to the physical configuration of the system.

The regression analysis results obtained using the FOPDT model, as shown in Figure 3, provide specific values of the process gain for each gas species. For ${\mathrm{CF}}_{3}$ and $O_{2}$, the process gains ($K_{CF_{3}}$ and $K_{O_{2}}$) are determined to be −27,060.9 and −7188.5, respectively, indicating that their concentrations decrease rapidly as the plasma voltage is applied. Conversely, the process gain ($K_{CO_{2}}$) for ${\mathrm{CO}}_{2}$ is 4785.6, indicating an increase in its concentration with the application of plasma voltage. These results suggest that fluorocarbons are rapidly oxidized in plasma environments.

The ToF-MS data were collected by controlling the process environment using the system described in Figure 1. To simulate reactions at both room temperature and the conditions under which ACL deposition and etching occur, eight repeated experiments were conducted at four temperature conditions: 20 °C, 200 °C, 400 °C, and 650 °C. During these experiments, the ICP plasma power was varied from 100 W to 900 W. A total of 288 datasets were obtained, which were used to calculate the process gain for F radicals, as well as for each gas species selected based on PCA importance. This calculation was performed using the FOPDT model. Figure 4a–k present these process gains as two-dimensional color maps, illustrating their dependence on process conditions. Additionally, the correlation between different gas species is visualized in Figure 4l.

A detailed analysis of the ${\mathrm{CF}}_{4}$ mixed plasma was conducted, covering the following elements: $O_{2}$, introduced into the process; ${\mathrm{CF}}_{3}$ and ${\mathrm{CF}}_{2}$, produced from the decomposition of ${\mathrm{CF}}_{4}$ [31]; the inert gas Ar; and the species F radicals—CO, ${\mathrm{CO}}_{2}$, COF, ${\mathrm{COF}}_{2}$, and ${\mathrm{COF}}_{3}$—formed by the plasma reactions. The concentration of F radicals, which play a crucial role in the etching process, generally increased with rising plasma power and gas temperature within the chamber. However, at a high temperature of 650 °C, the F-radical concentration peaked around 300 W and slightly decreased as plasma power increased further (Figure 4a). This decrease in F-radical concentration at higher power levels is likely due to the increased formation of carbonyl fluoride compounds (Figure 4i–k), which consume F radicals as they are generated.

At temperatures around 400 °C, with high plasma power applied, the formation of $C_{2}F_{5}$ (Figure 4d) can be explained by the decomposition of ${\mathrm{CF}}_{4}$ into lower fluorocarbon ions such as ${\mathrm{CF}}_{2}$, followed by their recombination to form larger fluorocarbon species [34,35]. This is consistent with our observation that the decrease in ${\mathrm{CF}}_{2}$ concentration is relatively small in the regions where $C_{2}F_{5}$ concentration increases, suggesting that $C_{2}F_{5}$ is formed when sufficient ${\mathrm{CF}}_{2}$ ions are available in the high-power plasma environment.

Additionally, Figure 4e shows that the concentration of Ar remained relatively unchanged under various plasma conditions, indicating that Ar does not participate in the plasma reactions. As the temperature rises, the concentration of $O_{2}$ decreases steadily (Figure 4f), while the concentrations of ${\mathrm{CO}}_{2}$ and CO show an upward trend (Figure 4g,h). This increase in CO concentration can be attributed to the behavior of COF intermediates within the ${\mathrm{CF}}_{4}$/$O_{2}$ plasma. ${\mathrm{CF}}_{2}$ radicals react with $O_{2}$ to form COF intermediates [36,37], which can either dissociate into CO or react further with $O_{2}$ to produce ${\mathrm{CO}}_{2}$. With increasing plasma power, the dissociation of COF into CO becomes more dominant compared to the formation of ${\mathrm{CO}}_{2}$, leading to a higher production of CO. This shift in reaction pathways, driven by elevated plasma power, results in a preferential generation of CO over ${\mathrm{CO}}_{2}$ as power increases, thereby explaining the observed rise in CO concentration [38].

However, at 650 °C and plasma power levels exceeding approximately 400 W, the concentrations of ${\mathrm{CO}}_{2}$ and CO begin to decrease again. This behavior suggests that at higher power levels, these species may be recombining to form other carbonyl fluoride compounds. The data revealed that as the energy input increased at higher temperatures and plasma power levels, the production of ${\mathrm{COF}}_{2}$ became more thermodynamically favorable [39].

Moreover, under high-temperature conditions (at 650 °C) and with plasma powers above 600 W, the concentration of ${\mathrm{COF}}_{3}$ increased significantly. This finding implies that the elevated energy levels facilitate the further fluorination of ${\mathrm{COF}}_{2}$, resulting in the generation of ${\mathrm{COF}}_{3}$. The additional energy supplied by the increased plasma power likely provides the necessary activation energy for these fluorination reactions, suggesting that ${\mathrm{COF}}_{3}$ is a secondary product formed when excess F radicals react with ${\mathrm{COF}}_{2}$ under these high-energy conditions.

Figure 4i visually represents the complex interactions within the entire process to illustrate the interdependencies between chemical species in the plasma. The left side shows the correlation matrix, where each cell represents the correlation coefficient between two chemical species. The color gradient indicates the correlation strength, with blue representing negative correlations and orange representing positive correlations.

The right side of Figure 4i displays a network map constructed from the correlation matrix. In this map, gas species that increase in concentration when the plasma is turned on (K > 0) are labeled in red, while those that decrease (K < 0) are labeled in blue. The font size of each label corresponds to the magnitude of concentration change. Connections between the species are depicted as lines, with the color and thickness of the lines indicating the strength and type of correlation: purple lines represent positive correlations, while orange lines denote negative correlations. In the network, the input gases ${\mathrm{CF}}_{3}$ and $O_{2}$ are fixed at the top-left corner, while the main product F is positioned at the bottom-right corner, providing a clear overview of the dominant interactions throughout the entire set of experimental conditions. This network visualization not only highlights the interdependencies and reaction pathways between the various chemical species in the plasma environment but also allows for a straightforward comparison of which species are predominantly involved in reactions under different conditions. In particular, the next section will provide insights into process control by clustering similar processes and analyzing each cluster comparatively.

### 4.4. Dimensionality Reduction and Clustering Analysis of Plasma Process Data

PCA is a powerful tool for reducing the dimensionality of complex datasets, enabling the identification of key patterns and relationships between variables that may not be immediately apparent. By transforming a large set of variables into a smaller set of principal components, PCA allows for the visualization of high-dimensional data in a lower-dimensional space, facilitating the interpretation of intricate interactions between process parameters and chemical species. In this study, PCA was employed to effectively visualize the complex interactions between process conditions and the resulting gas-phase species. The analysis was conducted using the process gain (K) values for each chemical species under various process conditions, and the resulting principal components were mapped onto a two-dimensional plane, as shown in Figure 5.

In Figure 5a, the process gains of gas species collected under varying temperature and plasma power conditions are visualized in a reduced-dimensional space defined by principal component 1 (PC1) and principal component 2 (PC2) through PCA. The color of each data point corresponds to the specific process condition, while its coordinates indicate the position within this reduced-dimensional space.

To better interpret this reduced-dimensional space, the original process gains (K) of each chemical species are represented as arrows in the lower-right corner of Figure 5a. These arrows are loading vectors representing the contribution of each chemical species to the principal component axes within the two-dimensional space. Because the scales of the *x*- and *y*-axes differ, dashed lines were used to represent vectors of equal length. The direction and length of each arrow show how the process gain for each gas species influences the position of data points in the reduced-dimensional space. Specifically, the direction of each arrow signifies the axis along which variations in a particular chemical species are most pronounced, while the length of the arrow reflects the extent of the species’ contribution to variations in the corresponding principal component. This enables a straightforward interpretation of the relationship between high-dimensional process variables and the low-dimensional representation highlighting how variations in the chemical species shape the distribution of experimental data.

By leveraging these loading vectors, Figure 5a effectively conveys the relationship between process conditions and the behavior of individual chemical species, providing a comprehensive overview of their interdependencies. This visualization underscores the dominant species influencing variance in the principal component space, thereby offering valuable insights into interaction pathways and reaction dynamics within the plasma process under different experimental conditions.

Figure 5b,c illustrate the loadings of each chemical species on the first and second principal components (PC1 and PC2), respectively. Positive or negative values indicate the direction and magnitude of the influence of each species on the principal components, thereby providing insight into which species drive variations along these key axes. The PC1 is defined as the linear combination of *K* that captures the maximum variance in the data. The PC2 is the linear combination that captures the second highest variance in the data, subject to being orthogonal to PC1. This orthogonality constraint ensures that PC2 captures a new, independent direction of variance that was not accounted for by PC1. As a result, this approach preserves as much information as possible while reducing its dimensionality, allowing for a clear visualization of complex relationships between variables in a simplified two-dimensional space.

Building on these PCA results, clustering was applied to the process data to further analyze patterns and relationships. PCA effectively reduces the dimensionality of high-dimensional data while preserving the most critical variance, thereby revealing inherent patterns and relationships among the variables. The k-means algorithm was used to cluster experimental conditions with similar process gains in the reduced-dimensional space, identifying four distinct groups based on the Elbow Method. As shown in Figure 6a, the PCA scatter plot displays these clusters, indicating underlying groupings in the chemical species’ behavior or process conditions. These groupings suggest different operational regimes or reaction pathways that are not easily observable in the original high-dimensional space.

Figure 6b–e provide a detailed analysis of each cluster, displaying the correlations among process variables. For each subfigure (b, c, d, and e), the left panel shows the correlation matrix for the gas species within the cluster, while the right panel presents a network map based on these correlations. The network map highlights strong correlations, making it easier to identify which species interact closely in each cluster.

In Cluster 1 (Figure 6b), the production of carbonyl fluoride compounds (e.g., COF, ${\mathrm{COF}}_{2}$) is lower compared to other clusters, with limited ${\mathrm{CF}}_{3}$ breakdown and reduced F-radical generation, suggesting fewer reactions leading to F radicals. In Cluster 2 (Figure 6c), COF, ${\mathrm{COF}}_{2}$, ${\mathrm{CF}}_{3}$, and $O_{2}$ show high correlations, with an increased ${\mathrm{CF}}_{3}$ breakdown and F-radical generation relative to Cluster 1, indicating that $O_{2}$ is promoting F-radical formation through its interaction with ${\mathrm{CF}}_{4}$, resulting in higher production of fluorinated compounds. In Cluster 3 (Figure 6d), the correlations between gas species become stronger, suggesting that the reactions observed in Cluster 2 are more pronounced here, with CO playing a more active role in forming COF and ${\mathrm{COF}}_{2}$ [36,37,39], leading to an increased production of carbonyl fluoride compounds. In Cluster 4 (Figure 6e), the correlation between carbonyl fluoride compounds and other species decreases relative to Cluster 3, with a significant drop in ${\mathrm{CF}}_{3}$ concentration accompanied by a notable increase in F-radical generation, indicating extensive ${\mathrm{CF}}_{3}$ decomposition and a corresponding rise in F radicals.

The methodology presented in this study effectively captures the complex interactions of plasma processes by combining PCA and clustering techniques. Through dimensionality reduction, we were able to highlight key patterns and visualize relationships between chemical species more clearly. Clustering in this reduced space allowed for the identification of distinct process behaviors, revealing variations in chemical reactions under different conditions. Additionally, there is the versatility to use different clustering methods to focus on specific aspects of the process, allowing for a more customized analysis of plasma chemistry.

Furthermore, the clustering results provide valuable insights for process control and optimization. By identifying how different process conditions influence reaction pathways within the plasma, we can determine which gas species are produced or consumed under specific conditions. This understanding enables us to adjust process parameters, such as temperature and plasma power, to achieve the desired plasma behaviors associated with specific clusters. For example, if the goal is to enhance the formation of beneficial chemical species like F radicals or to suppress the production of undesirable byproducts, we can modify the temperature and plasma power to operate within the conditions corresponding to the cluster that exhibits these characteristics. This approach offers concrete guidance on controlling process variables to achieve specific plasma behaviors, thereby improving the efficiency and outcomes of the plasma process.

## 5. Conclusions

In this study, we investigated high-temperature gas plasma environments used for ACL deposition and chamber cleaning processes. By utilizing a high-temperature ICP system coupled with ToF-MS, we analyzed variations in gas species under different plasma power and temperature conditions. Through the application of PCA and NMF, we identified the key gas species involved in these processes, while the FOPDT model provided an accurate quantification of dynamic changes in gas concentrations.

Our results show that the concentration of fluorine radicals and other gas species depends strongly on process parameters. For instance, we observed the formation of ${\mathrm{COF}}_{3}$ at high gas temperatures and plasma power levels, indicating the activation of new reaction pathways that are not present at lower temperatures. This finding underscores the importance of understanding how process parameters influence chemical reactions in high-temperature plasma environments. Additionally, visualization techniques such as dimensionality reduction and network maps helped illustrate these complex interactions more clearly, providing a deeper understanding of how different variables shape overall plasma behavior.

Our study provides the first in-depth experimental analysis of high-temperature gas plasma processes specifically tailored for semiconductor applications. While fluorocarbon plasmas are widely used in semiconductor manufacturing, their behavior at high gas temperatures has not been investigated until now. Our findings offer key insights that can be used to optimize plasma conditions and improve process control in high-temperature gas plasma environments.

## Figures and Tables

**Figure 1 sensors-24-07307-f001:**
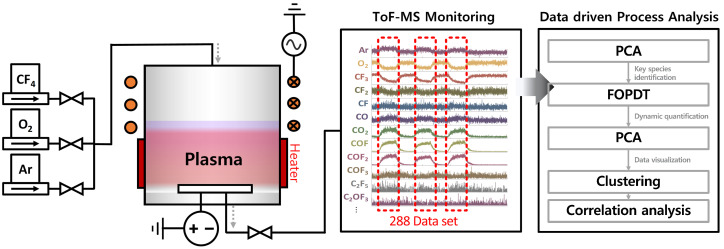
Schematic illustration of the experimental setup and data analysis workflow.

**Figure 2 sensors-24-07307-f002:**
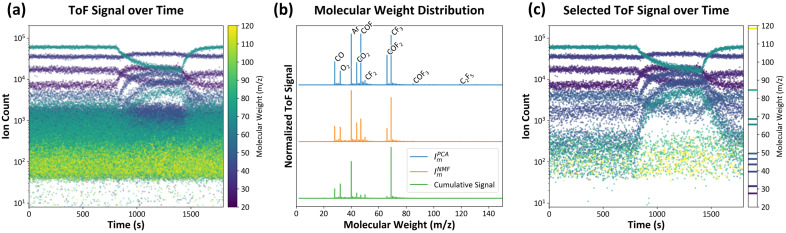
An analysis of the plasma environment was conducted using ToF-MS. The ICP plasma power was applied for 600 s at a process time of 800 s. (**a**) The number of ions detected by ToF-MS as a function of time is illustrated for molecular weights (*m*/*z*) ranging from 20 to 120. (**b**) The significance of the data is demonstrated through the use of PCA (blue), NMF (orange), and the cumulative signal over the entire time range (green). Ten major gas molecules were identified and labeled in the figure. (**c**) Only selected gas species are shown.

**Figure 3 sensors-24-07307-f003:**
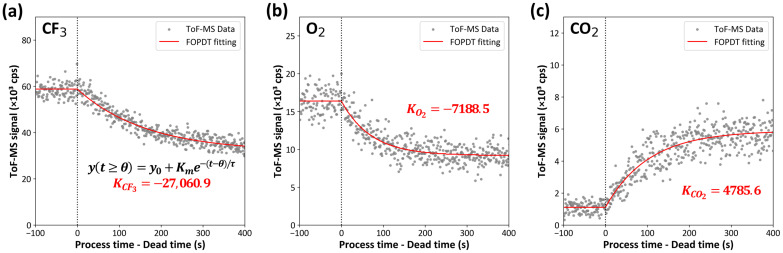
The time-dependent signal intensities of representative gas species were measured by ToF-MS during the process. The measured values of the input gases (**a**) ${\mathrm{CF}}_{3}$ and (**b**) $O_{2}$, and (**c**) ${\mathrm{CO}}_{2}$ produced by the plasma reaction, are illustrated as gray dots. For each signal, the process gain *K* for each gas species was obtained by applying an FOPDT model fitting, taking into account the dead time when the plasma is switched off, and is plotted on each graph. The red lines represent the FOPDT model fitting to the experimental data.

**Figure 4 sensors-24-07307-f004:**
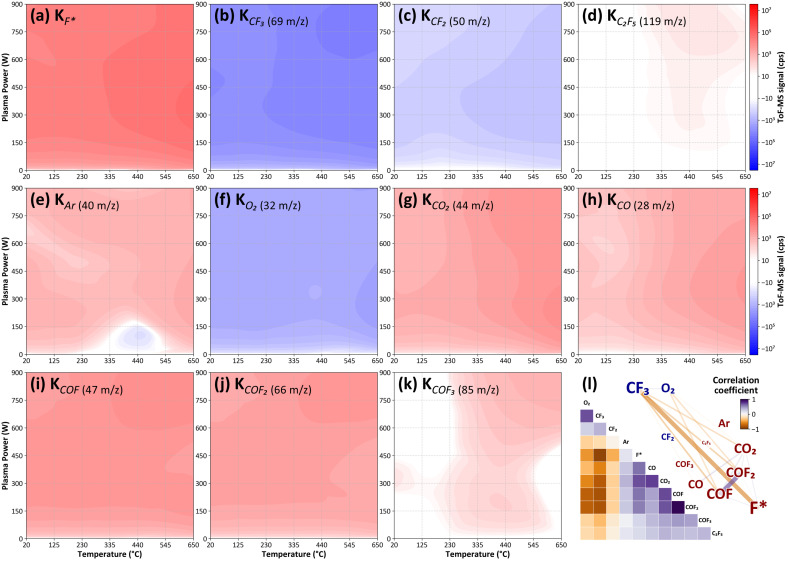
The process gains are plotted as two-dimensional color maps as a function of process gas temperature and plasma power using regression analysis. (**a**) The process gain for F radicals (F*) was calculated as the amount of gas participating in the reaction involving fluorine. (**b**–**k**) The process gains for other gas species selected according to their PCA importance are shown. Red areas indicate an increase in gas species when the plasma is turned on, while blue areas indicate a decrease. The color map is consistent across all figures. (**l**) The correlation matrix of gas species under all process conditions is shown, with higher correlation coefficients depicted in purple and lower coefficients in orange. On the right, a network map visualizes the interrelationships between the different gas species in the plasma environment, where increased species are shown in red and decreased species in blue, with font size representing the magnitude of the change.

**Figure 5 sensors-24-07307-f005:**
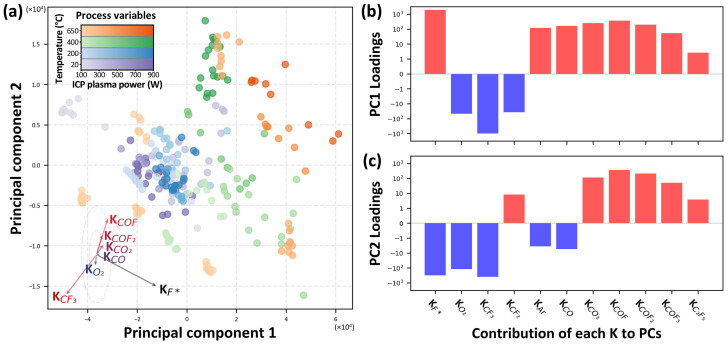
(**a**) Visualization of process gains (K) of gas species in a reduced-dimensional space defined by principal component 1 (PC1) and principal component 2 (PC2) using PCA. Each data point represents process conditions, colored according to temperature and plasma power values. The arrows in the lower left indicate the loading vectors of each gas species, showing their contribution to the principal components. (**b**,**c**) illustrate the loadings of each gas species on PC1 and PC2, respectively, indicating the direction and magnitude of their influence on the principal components. Red bars represent positive loadings, while blue bars indicate negative loadings, showing how each gas species contributes to variations along these key axes.

**Figure 6 sensors-24-07307-f006:**
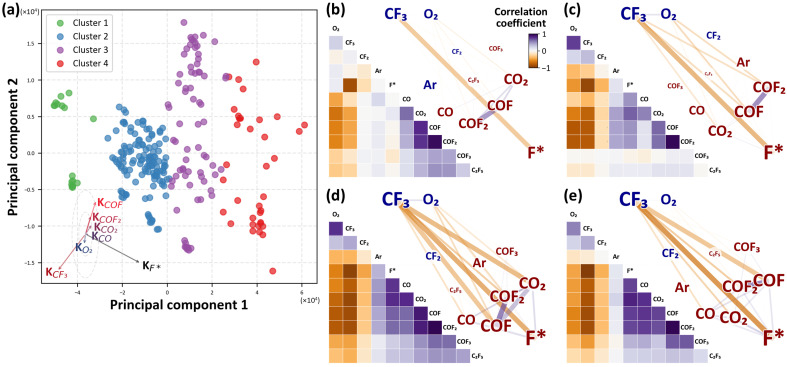
(**a**) Clustering of process conditions in the PCA-defined reduced-dimensional space, showing four clusters identified through k-means clustering. Each cluster represents similar behavior of gas species under different process conditions. The loading vectors of each gas species are shown as arrows to illustrate their contributions to principal component axes. (**b**–**e**) Correlation matrices and network maps for each cluster, visualizing the relationships between gas species. Positive correlations are shown in purple, while negative correlations are in orange. Increased species are marked in red, while decreased species are shown in blue, with the font size indicating the magnitude of the change.

**Table 1 sensors-24-07307-t001:** Comparison of relative importance values derived from PCA and NMF for different molecular weights. The table lists values for gas species with higher signals than their neighboring gas species, ordered by decreasing relative importance.

Molecular Weight	Relative PCA	Molecular Weight	Relative NMF
(*m*/*z*)	Importance $R_{m}^{\mathrm{PCA}}$ (%)	(*m*/*z*)	Importance $R_{m}^{\mathrm{NMF}}$ (%)
47	17.622963	40	22.174847
40	17.595561	69	19.153740
69	17.174095	47	9.838269
66	10.276480	44	8.105747
28	8.017336	28	6.678383
44	7.707956	32	6.303442
32	4.820177	66	5.680802
50	1.590005	50	1.970766
85	0.437264	36	0.542563
119	0.213417	85	0.276671
78	0.152130	78	0.212098
97	0.112599	20	0.133319
100	0.107691	81	0.117157
131	0.095920	119	0.106920
87	0.093103	87	0.075410

## Data Availability

The data that support the findings of this study are available from the corresponding author upon reasonable request.

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
