# Peer review of "Data-Driven Analysis of High-Temperature Fluorocarbon Plasma for Semiconductor Processing"

_sensors, 2024, doi:10.3390/s24227307_

Round 1
Reviewer 1 Report
Comments and Suggestions for Authors
The manuscript titled "Data-Driven Analysis of High-Temperature Fluorocarbon Plasma for Semiconductor Processing" provides a comprehensive exploration of fluorine-based plasma in semiconductor processing, with a focus on understanding the effects of high-temperature conditions using various analytical methods. The combination of Time-of-Flight Mass Spectrometry (ToF-MS) and multivariate techniques like PCA and NMF offers a valuable perspective on optimizing the plasma etching process.
Overall, the experimental setup and data analysis are well-documented, which supports the reliability of the findings. A few minor points of improvement can enhance the quality of this work and its influential impact on this field.
- The author insists that the relative importance values obtained from PCA and NMF are similar as indicated in Table 1. Considering the differences in the rank and the values, is it logical to conclude these are similar? The differences should be more critically discussed to validate the comparability of these results.
- Please elaborate on how the author identifies the most significant gas species by comparing the results from PCA and NMF in Section 4.1. Additional clarity on the reasoning behind the selection criteria for the most significant gas species is needed.
- Using dimensional reduction techniques, the process windows were categorized by clusters 1-4. In Figure 6, the author tried to demonstrate the significance of this clustering. How effective is this clustering in understanding the process? What kind of physical interpretation can be provided for this clustering? How does it contribute to a deeper understanding of plasma characteristics for varying process conditions?
- Some abbreviations were not used consistently. For example, PCA was repeatedly used, although the abbreviation was explained at the beginning of this article. Please ensure the consistent use of abbreviations throughout the manuscript for clarity.
- There are some typographical errors in the article that need correction. A thorough proofread is recommended to address these.
Author Response
Authors’ Reply to the Reviewers' Comments on:
Manuscript ID: sensors-3278846
Data-Driven Analysis of High-Temperature Fluorocarbon Plasma for Semiconductor Processing
By Sung Kyu Jang , Woosung Lee , Ga In Choi , Ji Hun Kim , Minji Kang , Seongho Kim , Jong Hyun Choi , Seul-Gi Kim , Seoung-Ki Lee , Hyeong-U Kim* , Hyeongkeun Kim*
We deeply appreciate the effort that the reviewers have taken in reviewing our manuscript. Changes have been carried out according to the comments and highlighted in red color in this revised version of the manuscript. We hope that our revisions properly addressed all the points of the comments.
Response to Comments of Reviewer #1
The manuscript titled "Data-Driven Analysis of High-Temperature Fluorocarbon Plasma for Semiconductor Processing" provides a comprehensive exploration of fluorine-based plasma in semiconductor processing, with a focus on understanding the effects of high-temperature conditions using various analytical methods. The combination of Time-of-Flight Mass Spectrometry (ToF-MS) and multivariate techniques like PCA and NMF offers a valuable perspective on optimizing the plasma etching process.
Overall, the experimental setup and data analysis are well-documented, which supports the reliability of the findings. A few minor points of improvement can enhance the quality of this work and its influential impact on this field.
Q1. The author insists that the relative importance values obtained from PCA and NMF are similar as indicated in Table 1. Considering the differences in the rank and the values, is it logical to conclude these are similar? The differences should be more critically discussed to validate the comparability of these results.
→A1) Thank you for your valuable question. You pointed out whether it is logical to conclude that the relative importance values obtained from PCA and NMF are similar in Table 1, considering the differences in their ranks and values. We will address this concern by considering the fundamental differences between PCA and NMF, specifically in the context of plasma process monitoring and plasma chemistry.
To qualitatively understand the differences between PCA and NMF, we can refer to feature extraction experiments in image processing [1]. In image analysis, PCA allows for negative representations, which means it can generate images that are essentially inverted versions of the original. This capability enables PCA to capture opposing features or contrasts within images. Analogously, in the plasma chemistry domain, PCA's ability to handle negative loadings allows it to effectively capture conflicting relationships between variables, such as the production and consumption of chemical species due to reverse or secondary reactions within the plasma chamber.
In contrast, NMF cannot represent negative values, so it effectively captures partial features of non-negative shapes, such as facial components in face images. Since our ToF-MS data are inherently non-negative, NMF is particularly suitable for capturing additive and localized features within the plasma chemistry data. However, because the components in NMF are not orthogonal, there can be redundancy or overlap among variables. This means that certain chemical species might be unnecessarily emphasized multiple times across different components.
Given these fundamental differences, it is natural for the ranks and magnitudes of the relative importance values obtained from PCA and NMF to differ. PCA captures the global variance structure and can represent negative correlations, which is valuable in identifying chemical species involved in both the production and consumption processes within the plasma. NMF, emphasizing non-negative and additive parts-based representations, focuses on capturing the dominant features present in the data, which can be especially effective for our non-negative ToF-MS measurements.
However, when we analyzed the data collected from the plasma chamber using both methods, we found that the top eight chemical species identified by both PCA and NMF were identical. Additionally, the 9th and 10th most important chemical species in PCA were ranked 10th and 14th in NMF, respectively. These results indicate a high degree of similarity between the two methods in identifying the key chemical species involved in the plasma processes, despite their methodological differences.
Thank you for the opportunity to discuss the differences in values obtained from PCA and NMF. Rather than simply stating that the results are similar, we will explain the characteristics and differences of each methodology and directly mention which chemical species are commonly identified as important by both methods. By doing so, we aim to address the reviewer's concerns and will revise the manuscript accordingly.
→ (P. 8, Line. 285) This table lists gas species with local maximum importance values—those having higher importance compared to their neighboring species—and orders them by descending relative importance. Although the relative importance values obtained from PCA and NMF are generally similar, some differences in the rankings and magnitudes are observed due to the distinct characteristics of the two methods.
PCA captures the global variance structure of the data and allows for both positive and negative loading values, enabling the representation of opposing relationships between variables. This characteristic is particularly valuable in plasma chemistry, as it allows PCA to capture conflicting relationships between variables, such as the production and consumption of chemical species due to secondary reactions within the plasma chamber. In contrast, NMF emphasizes non-negative, additive parts-based representations, focusing on capturing dominant features present in the data [30], which can be especially effective for the non-negative ToF-MS measurements. However, because the components in NMF are not orthogonal, there can be redundancy or overlap among variables, potentially leading to certain chemical species being unnecessarily emphasized multiple times across different components. Despite these methodological differences, the top eight chemical species identified by both methods were identical, and the 9th and 10th most important species in PCA were ranked 10th and 14th in NMF, respectively. This similarity indicates that both methods consistently identify the key gas species involved in the plasma processes.
Q2. Please elaborate on how the author identifies the most significant gas species by comparing the results from PCA and NMF in Section 4.1. Additional clarity on the reasoning behind the selection criteria for the most significant gas species is needed.
→A2) In our analysis, both PCA and NMF consistently identified the top eight gas species as the most significant, based on their relative importance values exceeding 0.1% in both methods. Specifically, we focused on species where the relative importance values were significant in both analyses.
- m/z 85 (COF₃): The mass-to-charge ratio of 85 corresponds to COF₃, a fluorinated compound. We selected this species because it contains fluorine, which is crucial in plasma chemistry. Notably, COF₃ showed a high correlation with COF₂ and COF at elevated temperatures, as depicted in Figure 6e. This correlation indicates that COF₃ participates in reactions at high temperatures.
- m/z 119 (Câ‚‚Fâ‚…): The mass-to-charge ratio of 119 likely represents Câ‚‚Fâ‚…. This species exhibited a weak negative correlation with the input gases (CFâ‚“, Oâ‚‚) and a positive correlation with other reaction products. This indicates that Câ‚‚Fâ‚… is formed within the plasma chamber through reactions of the input gases. Therefore, we included it in our analysis as it indicates active plasma chemistry generating new fluorinated species.
On the other hand:
- m/z 78 (Estimated as Câ‚‚Fâ‚‚O): Although this species showed relative importance values over 0.1% in both PCA and NMF, it did not display clear correlations with other species. Due to the lack of significant interaction patterns or correlations, we did not prioritize it in our analysis.
- m/z 87: This signal had relative importance values below 0.1% in both methods. Additionally, we could not identify any plausible chemical species composed of carbon, oxygen, and fluorine corresponding to this mass in the NIST Standard Reference Data. Therefore, we decided not to include it
We will include this explanation in the revised manuscript to enhance the clarity of our methodology and address the reviewer's request for additional reasoning behind our selection criteria.
→(P. 9, Line. 304) In selecting the most significant gas species, those with relative importance values exceeding 0.1% in both PCA and NMF analyses were considered. Specifically, the mass-to-charge ratio (m/z) of 85 corresponds to COF₃, a fluorinated compound containing fluorine, which is crucial in plasma chemistry. Notably, COF₃ showed a high correlation with COFâ‚‚ and COF at elevated temperatures, indicating that COF₃ participates in reactions at high temperatures. Similarly, the m/z of 119 likely represents Câ‚‚Fâ‚…. This species exhibited a weak negative correlation with the input gases (CFâ‚“, Oâ‚‚) and a positive correlation with other reaction products, suggesting that Câ‚‚Fâ‚… is formed within the plasma chamber through reactions of the input gases, indicating active plasma chemistry generating new fluorinated species. On the other hand, m/z 78 (estimated as Câ‚‚Fâ‚‚O) showed relative importance values above 0.1% in both PCA and NMF, but did not show a clear correlation with the other species, so we excluded it from the analysis. By combining the relative importance values from both methods with chemical relevance and correlation analysis, the ten gas species discussed in the manuscript were selected as the most significant for the study.
Q3. Using dimensional reduction techniques, the process windows were categorized by clusters 1-4. In Figure 6, the author tried to demonstrate the significance of this clustering. How effective is this clustering in understanding the process?
→A3.1) Thank you for your insightful question regarding the effectiveness of our clustering approach in understanding the plasma process. The clustering analysis using dimensional reduction techniques (specifically PCA followed by k-means clustering) allowed us to identify distinct groupings (Clusters 1-4) within the data that correspond to different reaction pathways in the plasma process. This method reduces the complexity of the data, making it easier to detect patterns and relationships that are not readily apparent in the high-dimensional space. By simplifying complex data, we were able to reveal natural groupings that correspond to different plasma behaviors under varying process conditions. Additionally, understanding which cluster a particular set of process conditions falls into enables us to better predict the behavior of the plasma and potentially detect anomalies, thereby enhancing process monitoring.
What kind of physical interpretation can be provided for this clustering?
→A3.2) Each cluster represents a set of process conditions that lead to specific plasma chemical behaviors (Figure 6):
Cluster 1: Characterized by lower plasma power and temperature, leading to minimal decomposition of CFâ‚„ and lower production of reactive species. The plasma reactions are less intense, and the generation of fluorine radicals is limited.
Cluster 2: Represents intermediate plasma power and temperature, where there is increased decomposition of CF₄ and higher production of reactive species such as CF₃, leading to more significant plasma reactions.
Cluster 3: Corresponds to higher plasma power and temperature, with further increased decomposition of CFâ‚„ and higher concentrations of reaction products like COF and COFâ‚‚, indicating more complex plasma chemistry.
Cluster 4: At the highest plasma power and temperature, the plasma chemistry is dominated by extensive decomposition and secondary reactions, leading to the formation of species like COF₃ and increased fluorine radical generation.
How does it contribute to a deeper understanding of plasma characteristics for varying process conditions?
→A3.3) Contributing to a deeper understanding of plasma characteristics, the clustering helps us reveal reaction pathways and informs process optimization. By identifying how different process conditions influence the reaction pathways within the plasma, we can determine which species are produced or consumed under specific conditions. This understanding allows us to tailor process parameters, such as temperature and plasma power, to achieve desired plasma behaviors associated with specific clusters. For example, if we aim to promote the formation of certain beneficial chemical species or suppress the generation of undesirable byproducts, we can adjust the temperature and plasma power to operate within the conditions corresponding to the cluster that exhibits these desired characteristics. This provides concrete guidance on controlling process variables to achieve specific plasma behaviors. By knowing which cluster leads to the optimal plasma characteristics, we can systematically adjust the process parameters to reach that cluster, thereby improving efficiency and outcomes in the plasma process.
In response to the feedback provided by the reviewer, the revised manuscript includes an additional statement indicating that the process can be controlled based on the aforementioned understanding.
→ (P. 16, Line. 542) Furthermore, the clustering results provide valuable insights for process control and optimization. By identifying how different process conditions influence reaction pathways within the plasma, we can determine which gas species are produced or consumed under specific conditions. This understanding enables us to adjust process parameters, such as temperature and plasma power, to achieve desired plasma behaviors associated with specific clusters. For example, if the goal is to enhance the formation of beneficial chemical species like F radicals or to suppress the production of undesirable byproducts, we can modify the temperature and plasma power to operate within the conditions corresponding to the cluster that exhibits these characteristics. This approach offers concrete guidance on controlling process variables to achieve specific plasma behaviors, thereby improving the efficiency and outcomes of the plasma process.
Q4. Some abbreviations were not used consistently. For example, PCA was repeatedly used, although the abbreviation was explained at the beginning of this article. Please ensure the consistent use of abbreviations throughout the manuscript for clarity.
→A4) Thank you for bringing this issue to our attention. We have reviewed the manuscript and noticed that some abbreviations (PCA, ICP, ToF-MS, and NMF) were not used consistently throughout the text. We have carefully revised the manuscript to ensure that all abbreviations are used consistently after their initial definition.
Q5. There are some typographical errors in the article that need correction. A thorough proofread is recommended to address these.
→A5) Thank you for bringing the typographical errors to our attention. We have thoroughly proofread the manuscript and corrected all identified errors throughout the article.

Reviewer 2 Report
Comments and Suggestions for Authors
Authors provide the possibility to use the powerful ICP discharge in fluorocarbons-containing gases for the amorphous carbon layers deposition and chamber cleaning for semiconductor processing technologies. The time-of-fight mass-spectrometry is used for the monitoring of gas particles dynamics.
The key conclusion is that the macroscopic ICP parameters include the input power thus gas temperature are important so has to be thoroughly optimized and controlled.
The results are of interest for specialist in the area of semiconductor engineering.
The manuscript is well prepared and clearly illustrated so can be published in Sensors Journal.
Author Response
Authors’ Reply to the Reviewers' Comments on:
Manuscript ID: sensors-3278846
Data-Driven Analysis of High-Temperature Fluorocarbon Plasma for Semiconductor Processing
By Sung Kyu Jang , Woosung Lee , Ga In Choi , Ji Hun Kim , Minji Kang , Seongho Kim , Jong Hyun Choi , Seul-Gi Kim , Seoung-Ki Lee , Hyeong-U Kim* , Hyeongkeun Kim*
We deeply appreciate the effort that the reviewers have taken in reviewing our manuscript. Changes have been carried out according to the comments and highlighted in red color in this revised version of the manuscript. We hope that our revisions properly addressed all the points of the comments.
Response to Comments of Reviewer #2
Authors provide the possibility to use the powerful ICP discharge in fluorocarbons-containing gases for the amorphous carbon layers deposition and chamber cleaning for semiconductor processing technologies. The time-of-fight mass-spectrometry is used for the monitoring of gas particles dynamics.
The key conclusion is that the macroscopic ICP parameters include the input power thus gas temperature are important so has to be thoroughly optimized and controlled.
The results are of interest for specialist in the area of semiconductor engineering.
The manuscript is well prepared and clearly illustrated so can be published in Sensors Journal.
→ Thank you for your favorable review of this manuscript. The text has been revised to incorporate feedback from other reviewers to further enhance its completeness. Thank you again.

Reviewer 3 Report
Comments and Suggestions for Authors
1. The probe fixturing in the heater chamber used for ToF-MS monitoring should be explained further.
2. A flow chart may be added to explain the process of applying the FOPDT model to analyze the dynamic behavior of gas species signal intensities, and the whole process proposed in this manuscript.
Author Response
Authors’ Reply to the Reviewers' Comments on:
Manuscript ID: sensors-3278846
Data-Driven Analysis of High-Temperature Fluorocarbon Plasma for Semiconductor Processing
By Sung Kyu Jang , Woosung Lee , Ga In Choi , Ji Hun Kim , Minji Kang , Seongho Kim , Jong Hyun Choi , Seul-Gi Kim , Seoung-Ki Lee , Hyeong-U Kim* , Hyeongkeun Kim*
We deeply appreciate the effort that the reviewers have taken in reviewing our manuscript. Changes have been carried out according to the comments and highlighted in red color in this revised version of the manuscript. We hope that our revisions properly addressed all the points of the comments.
Response to Comments of Reviewer #3
Q1. The probe fixturing in the heater chamber used for ToF-MS monitoring should be explained further.
→ A1) Thank you for pointing out the need for further explanation regarding the gas sampling probe in the heater chamber used for ToF-MS monitoring. In the revised manuscript, we have included a more detailed description of how the ToF-MS probe is installed within the high-temperature heater chamber.
→ (P. 3, Line. 94) To ensure the formation of plasma at the appropriate position within the chamber, an electrically grounded chuck was installed inside the heater via vacuum feedthroughs. Downstream of the chamber, a gas collection line branched off from the main pump line and connected to the ToF-MS system. This gas sampling line was maintained at 100°C using heating tape to prevent condensation or deposition of gas species. Inside the ToF-MS, dry pumps were utilized to draw in the process gas from the chamber.
(Specifically, we have elaborated on the design of the sampling interface, the materials used to withstand high temperatures, and how the probe maintains a seal to ensure accurate gas sampling without compromising the chamber's environment.)
Q2. A flow chart may be added to explain the process of applying the FOPDT model to analyze the dynamic behavior of gas species signal intensities, and the whole process proposed in this manuscript.
→A2) We appreciate your suggestion to include a flow chart to explain the overall methodology proposed in our manuscript. In response, we have revised Figure 1 to incorporate a flow chart that visually outlines the entire experimental and analytical process.
→ (Figure 1)

Reviewer 4 Report
Comments and Suggestions for Authors
The authors applied Principal Component Analysis (PCA) to focus on the variations in fluorine-containing species to estimate the F radical concentration. The authors developed a high-temperature inductively coupled plasma system integrated with ToF-MS.
This research is interesting and novelty. This manuscript can be accepted after moderate revision.
The reference format is awful. References should be corrected as the journal format.
The authors should give more explanation on the optimizing Amorphous Carbon Layer processes in semiconductor manufacturing.
Author Response
Authors’ Reply to the Reviewers' Comments on:
Manuscript ID: sensors-3278846
Data-Driven Analysis of High-Temperature Fluorocarbon Plasma for Semiconductor Processing
By Sung Kyu Jang, Woosung Lee , Ga In Choi , Ji Hun Kim , Minji Kang , Seongho Kim , Jong Hyun Choi , Seul-Gi Kim , Seoung-Ki Lee , Hyeong-U Kim* , Hyeongkeun Kim*
We deeply appreciate the effort that the reviewers have taken in reviewing our manuscript. Changes have been carried out according to the comments and highlighted in red color in this revised version of the manuscript. We hope that our revisions properly addressed all the points of the comments.
Response to Comments of Reviewer #4
The authors applied Principal Component Analysis (PCA) to focus on the variations in fluorine-containing species to estimate the F radical concentration. The authors developed a high-temperature inductively coupled plasma system integrated with ToF-MS.
This research is interesting and novelty. This manuscript can be accepted after moderate revision.
The reference format is awful. References should be corrected as the journal format.
→ A) Thank you for bringing to our attention the inconsistencies in the reference formatting. We apologize for the oversight. We have thoroughly revised all references to conform to the journal's guidelines.
The authors should give more explanation on the optimizing Amorphous Carbon Layer processes in semiconductor manufacturing.
→ A) Thank you for suggesting that we provide more explanation on optimizing Amorphous Carbon Layer (ACL) processes in semiconductor manufacturing. In the revised manuscript, we have expanded the discussion on how our findings contribute to the optimization of plasma processes. We believe this will allow for improved control of the cleaning and deposition processes involved in the ACL process.
→ (P. 16, Line. 542) Furthermore, the clustering results provide valuable insights for process control and optimization. By identifying how different process conditions influence reaction pathways within the plasma, we can determine which gas species are produced or consumed under specific conditions. This understanding enables us to adjust process parameters, such as temperature and plasma power, to achieve desired plasma behaviors associated with specific clusters. For example, if the goal is to enhance the formation of beneficial chemical species like F radicals or to suppress the production of undesirable byproducts, we can modify the temperature and plasma power to operate within the conditions corresponding to the cluster that exhibits these characteristics. This approach offers concrete guidance on controlling process variables to achieve specific plasma behaviors, thereby improving the efficiency and outcomes of the plasma process.
